# Mutual Interference of Local Gravity Wave Forcings in the Stratosphere

**Nadja Samtleben [1],\*, Aleš Kuchař [1], Petr Šácha [2,3], Petr Pišoft [3] and Christoph Jacobi [1]**

[1]   Institute for Meteorology, Universität Leipzig, Stephanstr. 3, 04103 Leipzig, Germany;
      ales.kuchar@uni-leipzig.de (A.K.); jacobi@uni-leipzig.de (C.J.)

[2]   Institute for Meteorology, Universität für Bodenkultur Wien, Gregor-Mendel-Straße 33, 1180 Vienna, Austria;
      o.o.d.sachta@seznam.cz

[3]   Department of Atmospheric Physics, Faculty of Mathematics and Physics, Charles University,
      V Holesovickach 2, 180 00 Prague, Czech Republic; petr.pisoft@mff.cuni.cz

\*    Correspondence: Nadja.Samtleben@uni-leipzig.de

**Abstract:**   Gravity wave (GW) breaking and associated GW drag is not uniformly distributed among latitudes and longitudes. In particular, regions of enhanced GW breaking, so-called GW hotspots, have been identified, major Northern Hemisphere examples being located above the Rocky Mountains, the Himalayas and the East Asian region. These hotspots influence the middle atmosphere circulation both individually and in combination. Their interference is here examined by performing simulations including (i) the respective single GW hotspots, (ii) two GW hotspots, and (iii) all three GW hotspots with a simplified global circulation model. The combined GW hotspots lead to a modification of the polar vortex in connection with a zonal mean flow decrease and an increase of the temperature at higher latitudes. The different combinations of GW hotspots mainly prevent the stationary planetary wave (SPW) 1 from propagating upward at midlatitudes leading to a decrease in energy and momentum transfer in the middle atmosphere caused by breaking SPW 1, and in turn to an acceleration of the zonal mean flow at lower latitudes. In contrast, the GW hotspot above the Rocky Mountains alone causes an increase in SPW 1 amplitude and Eliassen–Palm flux (EP flux), inducing enhanced negative EP divergence, decelerating the zonal mean flow at higher latitudes. Consequently, none of the combinations of different GW hotspots is comparable to the impact of the Rocky Mountains GW hotspot alone. The reason is that the GW hotspots mostly interfere nonlinearly. Depending on the longitudinal distance between two GW hotspots, the interference between the combined Rocky Mountains and East Asian GW hotspots is more additive than the interference between the combined Rocky Mountains and Himalaya GW hotspots. While the Rocky Mountains and the East Asian GW hotspots are longitudinally displaced by 105°, the Rocky Mountains are shifted by 170° to the Himalayas. Moreover, while the East Asian and the Himalayas are located side by side, the interference between these GW hotspots is the most nonlinear because they are latitudinally displaced by 20°. In general, the SPW activity, e.g., represented in SPW amplitudes, EP flux or Plumb flux, is strongly reduced, when the GW hotspots are interacting with each other. Thus, the interfering GW hotspots mostly have a destructive effect on SPW propagation and generation.

**Keywords:** gravity waves; planetary waves; polar vortex; stratospheric warming

## 1. Introduction

The winter Northern Hemisphere (NH) stratosphere is mainly dominated by a strong west wind regime, the polar vortex, induced by a strong cooling. During the last decades the variability of the polar vortex induced by different circulation patterns, such as the Atlantic Oscillation (AO) [1–4] or

El Nino Southern Oscillation (ENSO) [5–9] connected with differences in the planetary wave (PW) activity [10,11] was analyzed in detail. Despite the fact that the impact of these atmospheric phenomena can be extracted from observations and can be qualitatively reproduced in climate models, there are still open questions. For example, frequently the simulated polar vortex is stronger than the observed one, and breaks up later in spring, and consequently model climatologies show a cold bias in the polar region in the Southern Hemisphere (SH) [12].

One of the dynamical processes that influence the evolution of the polar vortex are gravity waves (GWs) [13–17]. GWs are one of the main contributors to the middle atmosphere circulation by transporting and transferring energy and momentum throughout the whole atmosphere [18,19]. They are mainly generated by orography [20,21], convection [22], seismic activity [23,24] and spontaneous adjustment processes [18]. According to spatial distribution of the relevance of these mechanisms, GWs are not uniformly distributed over the globe or along latitudes, which is immediately seen from observations on a global scale [25–27], while long-term observations revealed even different trends in GW activity depending on longitude [28]. In particular, there are regions of enhanced GW activity in connection with increased momentum transfer, strongly influencing the atmospheric dynamics. While the most prominent regions of orographic GW generation at midlatitudes are the Rocky Mountains and the Himalayas provided that GWs are also generated owing to seismic activity near the Himalayas, regions of enhanced non-orographic GW activity are not locally fixed, owing to their temporal and spatial variability highly depending on the synoptic conditions [29–31].

Owing to the coarse resolution of global circulation models (GCMs), small-scale processes, such as GWs have to be parameterized, which may lead to biases if they are not represented correctly. However, most of the current orographic GW (OGW) parameterizations are able to reasonably capture the GW hotspots and the related momentum transfer [32]. Cohen et al. [33,34] showed that perturbations to the parameterized orographic GW forcing are partly compensated by resolved wave drag. Sigmond and Shepherd [35] confirmed the existence of the compensation mechansim between resolved Rossby wave drag and parameterized OGW drag in a comprehensive GCM. They argued that this mechanism needs to be considered in climate projections, especially for changes in the Brewer-Dobson circulation. However, Eichinger et al. [36] recently showed that the impact on the transport within the models cannot be compensated, which reinforces the necessity to study the complex OGW drag impact in comprehensive models.

Several studies already showed that the development of the polar vortex is strongly influenced if some GW parameters such as the GW drag (GWD) or the momentum flux are modified. E.g., Scheffler et al. [14] and Samtleben et al. [16,17] adjusted the GWD distribution guided by observations, and caused changes in the geometry and stability of the polar vortex. The induced weakening along with the deceleration of the zonal mean flow and the increase of the temperature destabilizes the polar vortex, which may reduce the cold bias in polar regions in model climatologies [37].

The forcing originating from a modified GWD distribution may act towards a preconditioning of the winter stratosphere for sudden stratospheric warmings (SSWs), emphasizing that GW strongly contribute to the polar vortex dynamics in addition to the PWs. Garcia et al. [12], after increasing the orographic GW flux in the WACCM GCM, observed a strengthening of the Brewer-Dobson circulation leading to a warmer polar region in the SH. As a result of the enhanced GW forcing, the polar vortex conditions were comparable to those obtained from MERRA reanalysis data. The same approach was applied by Polichtchouk et al. [38], but they were focusing on parameterized non-orographic GWs. They found that the number of SSWs in the SH increases and that the polar vortex is less persistent, when the GWD is enhanced. Ren et al. [39] removed the topographical forcing of the Rocky Mountains and the Himalaya in the WACCM GCM, and found that both regional forcings lead to a warming of the polar stratosphere and to a displacement of the polar vortex. They also observed that the combined effect from both regions on the polar vortex is weaker than the influence of one region alone, because the topographic forcings mutually dampen the dynamical changes like wave propagation.

Šácha et al. [26] and Adushkin et al. [23] have highlighted the importance of another GW hotspot in the East Asian region, and it has been shown that this hotspot may have considerable influence on the polar vortex [13,16]. Therefore, extending the experiments by Ren et al. [39], we here analyze the interference of breaking GW hotspots near the Rocky Mountains, the Himalayas and the observed East Asian hotspot, and their impact on the circulation in the middle atmosphere. In Section 2, we introduce the used mechanistic GCM, the Middle and Upper Atmosphere Model (MUAM) and describe the implementation of the local GW forcings. Section 3 describes and discusses the effects of the local GW forcings on the atmospheric dynamics including the background conditions, the PW activity and the polar vortex stability. Section 4 briefly summarizes the paper.

## 2. Numerical Model Experiments

### 2.1. Model Description

To simulate the effect of localized GW forcings on the polar vortex dynamics during NH winter, we performed perpetual January simulations. Therefore, we used the Middle and Upper Atmosphere Model (MUAM) [16,17,27,40,41], a three-dimensional, mechanistic, nonlinear global circulation model solving the primitive equations [42]. It has a horizontal resolution of $5°$ in latitude and $5.625°$ in longitude and extends in 56 layers up to an altitude of about 160 km in logarithmic pressure height. Throughout the lower and middle atmosphere the logarithmic pressure height mostly corresponds to the geometric height with small deviations of less than 5 km [40]. At 1000 hPa, the lower boundary of the model, the zonal mean temperature and geopotential height as well as the stationary planetary waves (SPWs) with wavenumbers 1–3 extracted from ERA-Interim January 2000–2010 temperature and geopotential height reanalysis data [43]. To correctly reproduce the dynamics in the troposphere, we nudged the model zonal mean temperatures to 2000–2010 mean ERA-Interim zonal mean temperatures up to an altitude of 10 km. As for the previous experiments [16,17] we choose the decadal mean from 2000–2010 because single years can strongly differ among each other with respect to the stability of the polar vortex in connection with the PW activity. The decadal average includes years with a strong as well as with a weak polar vortex and different PW characteristics, i.e., we do not analyse any extremes. The $H_2O$, $CO_2$ and $O_3$ fields, which are prescribed, are relevant for the solar and infrared radiative processes parameterized according to Strobel [44], Fomichev and Shved [45] and Fomichev et al. [46]. The parameterizations include the emission and absorption properties of the prescribed atmospheric constituents as well as additional absorption bands like the extreme ultra violet (EUV) band in the thermosphere. Other thermospheric effects like Rayleigh friction, Newton cooling and ion drag are also parameterized.

To represent the impact of atmospheric GWs within MUAM, an updated Lindzen-type linear scheme [42,47] with multiple breaking levels [48,49] is used. Once a GW becomes unstable, the amplitude saturates and stops increasing exponentially with altitude. The GW momentum flux is simultaneously reduced while the zonal mean flow is de- or accelerated. The GW can have multiple saturation levels until the GW encounters a critical line (GW phase speed is equal to the zonal wind) and the momentum flux becomes equal to zero [50]. At each grid point at 10 km altitude, 48 GWs are initialized propagating into eight different directions with six different phase speeds from $5\,\mathrm{ms}^{-1}$ up to $30\,\mathrm{ms}^{-1}$ in steps of $5\,\mathrm{ms}^{-1}$. Thus, the model only considers GWs having non-zero phase speeds and does not represent stationary mountain waves. The zonal mean GW amplitudes are initiated at 10 km altitude with a global mean vertical velocity perturbation of $1\,\mathrm{cm\,s}^{-1}$. This value is weighted with a prespecified global GW amplitude distribution based on GW potential energy data derived from Global Positioning System (GPS) radio occultation measurements [16,17,26]. This approach may lead to potential biases in the equatorial region because other waves with similar vertical wavelength such as Kelvin waves were not filtered out.

## 2.2. Reference MUAM Run

In the described model configuration we performed a reference (Ref) simulation with a spin-up period of 270 model days, during which several atmospheric waves such as SPWs and tides are switched on and the mean circulation develops. Simulations are performed for perpetual January conditions. To avoid further fluctuations we fixed the declination corresponding to 15 January, the middle of the analysed month. The ozone and carbon dioxide concentrations are taken as constant, they are based on the data from year 2005, which represents the mid of the decade 2000–2010. Our experiment additionally involves 120 model days with a temporal resolution of 2 h. At large, for each experiment the model is running for 390 model days, from which we mostly analyse the last 30 model days. These 30 days represent a monthly mean state of the atmosphere when calculating the average because (i) we only use monthly mean boundary conditions and (ii) the model reaches almost a steady state with small fluctuations after the implementation of the GW forcing. Because the Ref simulation is identical to the reference simulation shown by Samtleben et al. [16], in Figure 1 we only briefly present the zonal mean zonal wind, GW acceleration induced by breaking GWs and the GW flux, and the SPW 1 amplitudes and phases, for a discussion of the model performance regarding the background circulation. The model reproduces well the zonal mean zonal wind (a) and temperature (d) distributions in comparison, e.g., to the CIRA-86 [51] or URAP [52] climatologies. However, the stratospheric jet maximum southward of $60°$ N and above $50\,\mathrm{km}$, exceeding $60\ \mathrm{ms^{-1}}$, is somewhat overestimated by about $10$–$20\ \mathrm{ms^{-1}}$ in relation to the CIRA-86 and URAP climatologies. Nevertheless, comparisons to zonal mean zonal wind time series averaged between 60 and $80°$ N from satellite observations [53] show that the strength of the mesospheric jet is realistic. The overestimation of the zonal wind is mainly induced by the underestimation of the SPW 1 amplitudes in connection with a less intense momentum transfer and therefore, a too weak deceleration of the polar vortex [54]. But nevertheless, the distribution of the SPW 1 phase and SPW 1 amplitudes with maximum amplitudes of $10\,\mathrm{K}$ around $60°$ N are comparable to observations but the latter are less pronounced [55]. The SPW 1 amplitudes were calculated with the help of a harmonic analysis based on the method of the smallest error squared.

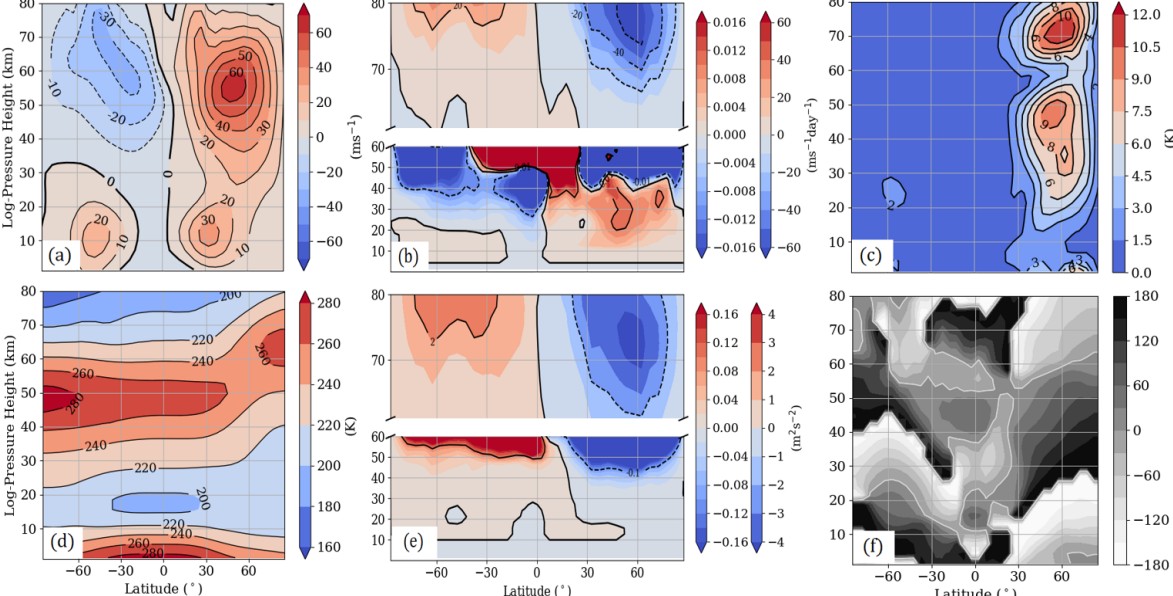

**Figure 1.** Zonal mean monthly mean (**a**) zonal wind ($\mathrm{ms^{-1}}$), (**b**) gravity wave (GW) zonal wind acceleration ($\mathrm{ms^{-1}day^{-1}}$) with two different scalings (left color bar-up to $60\,\mathrm{km}$, right color bar-above $60\,\mathrm{km}$), (**c**) SPW 1 amplitude (K), (**d**) temperature (K), (**e**) zonal GW flux ($\mathrm{m^2s^{-2}}$) with two different scalings (left color bar-up to $60\,\mathrm{km}$, right color bar-above $60\,\mathrm{km}$) and (**f**) stationary planetary wave (SPW) 1 phase ($°$). Results refer to January conditions, and to the reference (Ref) simulation.

This approach considers each parameter (e.g., the temperature) to be a superposition of cosine and sine functions. Thus, the background state of the atmosphere is normally added up with the harmonic components depending on the wavenumber and the longitude. According to linear GW theory, we observe a negative (positive) GWD (acceleration induced by breaking GWs) of about $-60\,\text{ms}^{-1}\text{day}^{-1}$ ($20\,\text{ms}^{-1}\text{day}^{-1}$) and GW flux of about $-4\ \text{m}^2\text{s}^{-2}$ ($2\,\text{m}^2\text{s}^{-2}$) in the NH (SH), which is comparable to observations [53].

*2.3. Experiment Description*

Following the strategy applied in the numerical experiments by Samtleben et al. [16], we now include an additional GW forcing by locally increasing the GWD after the spin-up period. For our experiments we focus on two regions mainly influenced by orographic GWs: the Rocky Mountains (RM) and the Himalayas (HI). We analyse their effect on the stratospheric circulation, and their interactions with the observed GW breaking hotspot in the East Asian/North-West Pacific region (EA) [13]. The chosen intensity of the forcing in each hotspot as well as the size of the simulated GW hotspots is the same in each experiment, in order to make their effects quantitatively comparable.

To modify the GW forcing, the zonal ($\text{GWD}_u$) and meridional ($\text{GWD}_v$) GWD as well as the heating ($\text{GWD}_T$) due to breaking GWs were adjusted. To this end, the original GW parameterization routine could not be used any more. Because the enhanced GW forcing leads to changes in the circulation, which in turn modulate the GW breaking and propagation conditions and thus, the GWD and its distribution, we turned off the GW parameterization in the experiments with modified GWD distribution. Instead, we used the GWD fields from the Ref simulation, enhanced the GWD in the respective hotspot regions, and forced the model with the modified GWD fields. According to Šácha et al. [13] the EA hotspot was located between 37.5° N–62.5° N and 118.1° E–174.3° E in an altitude range between 18 and 30 km. With the same latitudinal (25°), longitudinal (56.25°) and altitudinal (12 km) extent the size of the RM and HI GW hotspots was chosen. In our experiments, the RM GW hotspot was located between 27.5° N–52.5° N and 78.7° W–135° W, and the HI hotspot was located between 22.5° N–47.5° N and 61.8° E–118.1° E. The positions of the RM (cyan), HI (white) and EA (blue) GW hotspots as well as the geopotential height of the Ref simulation averaged between 20 and 30 km are shown in Figure 2. With respect to the size and the three-dimensional box-like shape of the GW hotspots, we also performed additional MUAM experiments including (i) a larger or smaller EA GW hotspot or (ii) a three-dimensional Gaussian distribution for the EA GW hotspot to see if the results strongly differ. Changes in the vertical or horizontal extension of the EA GW hotspot show that the zonal mean zonal wind, e.g., differs by about $3\,\text{ms}^{-1}$ in maximum. The comparison of the SPW 1 amplitudes between the box-like shape and the Gaussian distribution shows that the amplitudes may vary by about $4\,\text{ms}^{-1}$ in maximum (see Figure 8 in Samtleben et al. [16]).

In an earlier study [13] the intensity of the GW forcing was varied and the response of the middle atmosphere was analysed. Larger values led to a vortex breakdown and to a rearrangement of the whole circulation, while smaller values only caused small deviations from the reference circulation. For our experiments we choose values leading to moderate effects on the atmospheric dynamics, i.e., a $\text{GWD}_u$ with about $-10\,\text{ms}^{-1}\text{day}^{-1}$, a $\text{GWD}_v$ with about $-0.1\,\text{ms}^{-1}\text{day}^{-1}$ and a $\text{GWD}_T$ with about $0.05\,\text{Kday}^{-1}$. Thus, the maximum $\text{GWD}_u$ in the region of the EA GW hotspot is 500 times larger than the one of the Ref simulation. The modified maximum $\text{GWD}_v$ and $\text{GWD}_T$ is only 5 times larger. An increase of the zonal GWD by a factor of 500 represents a strong modification, however, observations and also other GW parameterizations show GWD values of even more than $-40\,\text{ms}^{-1}\text{day}^{-1}$ [32]. To analyse the effect of each GW hotspot and the interference of these different hotspots, we performed experiments with single GW hotspots (RM, HI, and EA), with two hotspots (RM + HI, RM + EA, HI + EA) and with all three GW hotspots (RM + HI + EA). Clearly, adding a GWD hotspot means that the overall GWD forcing is enhanced. This is the more the case with the experiments including two hotspots and the one with three hotspots. This approach also allows us to indirectly analyse, whether the mean circulation linearly responds to localized forcing, or not.

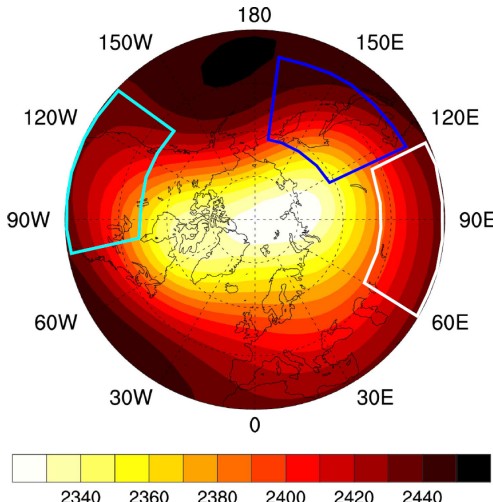

**Figure 2.** Geopotential height of the Ref simulation averaged between 20 and 30 km northward of 30°. Positions of the respective GW hotspots. Blue box—East Asia (EA); white box—Himalayas (HI); cyan box—Rocky Mountains (RM).

From our previous studies [16,17] we already know that the impact of single artificial GW hotspots is strongly depending on the modeled SPW phase. Observations from 2002 to 2007 [56] show that the SPW phase varies interannually during the winter season, which may be partly caused by strong North Atlantic Oscillation (NAO) or ENSO events but also by the Quasi-Biennial Oscillation (QBO) and SSW. If we would use single years as boundary conditions for the model, we would observe different effects of the localized GW forcings, which can be traced back to the deviations in the SPW phase distribution. This assumption was proven by performing experiments including boundary conditions of years with strong NAO and ENSO events (not shown here). In our approach we do not want to include these extreme events and used the decadal mean as boundary conditions. Thus, our additional GW forcing does not influence a polar vortex, which is not already strongly stabilized or destabilized by different meteorological phenomenon in connection with a strong or weak PW activity.

## 3. Effect on the Circulation

In this paragraph, we will describe the effects of combined GW hotspots on the mean circulation. The positions of the RM and HI GW hotspots are comparable (although not the same) to some of those GW hotspots analyzed by Samtleben et al. [17]. Thus, the background changes caused by the single GW hotspots are expected to be similar to the results presented by Samtleben et al. [17] and will not be discussed in this section. However, the figures include all experiments, both single ones and in combination to be able to relate the impact of the combined GW hotspots to the single GW hotspots.

### 3.1. Hotspot Effects on the Background Circulation

Figure 3 shows the zonal mean temperature differences (a) and zonal wind differences (b) between the single GW hotspots (cyan, blue and black contour lines) and the Ref simulation as well as for the doubled GW hotspots (red, orange and grey contour lines) and the tripled GW hotspot (yellow contour line) averaged between 18 and 30 km (temperature) and 18 and 60 km (zonal wind). The altitude range for the temperature differences is smaller because of the cooling occuring above 30–40 km (see Figure A1), which would average out the warming. In general, the different combinations of all three GW hotspots lead to a warming of the polar stratosphere in combination with a decreasing zonal mean zonal wind at middle and higher latitudes. Thus, the additional GW forcings lead to a weakening of the polar vortex, which can be interpreted as a modification of the polar vortex that may result in a SSW.

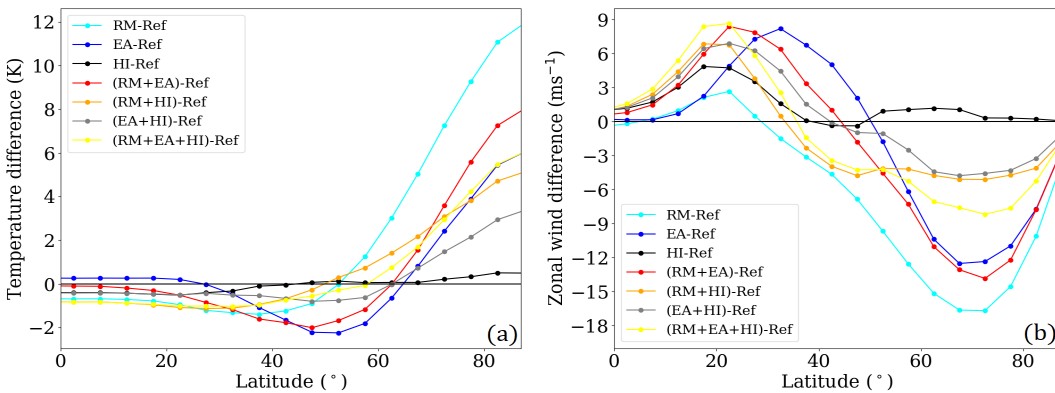

**Figure 3.** Zonal mean temperature differences (**a**) and zonal wind differences (**b**) between the single GW hotspots, the doubled GW hotspots and the tripled GW hotspots and the Ref simulation averaged between 18 and 30 km.

The zonal mean zonal wind is increasing at lower latitudes indicating the displacement of the polar vortex towards lower latitudes (underlined by Figure A2). The combination of the RM and EA GW hotspots (Figure 3b—red contour line) initiates a deceleration of the zonal mean flow of up to $-13\,\mathrm{ms}^{-1}$, which is weaker (stronger) than the one for the RM (EA) simulation. However, summing up the differences of RM-Ref and EA-Ref would lead to a total difference of up to about $-26\,\mathrm{ms}^{-1}$ in maximum. Thus, the interference of both GW hotspots leads to a zonal wind deceleration that is roughly half as strong as expected from a purely linear effect. In contrast to the wind anomalies at higher latitudes, the acceleration at lower latitudes southward of $30°$ N has intensified and is stronger than in both the RM and the EA simulation. The sum of the respective single GW hotspot positive zonal wind differences are similar or partly smaller than the positive zonal wind differences induced by the combined GW hotspots. As well for the temperature, the strength of the positive anomalies in the polar region in (Figure 3a—red contour line) are weaker (stronger) than those of the RM (EA) experiments. Thus, we observe (i) a destructive interference of the effects at middle and higher latitudes leading to smaller zonal wind and temperature anomalies as expected from linear interference of both single GW hotspots and (ii) a constructive interference at lower latitudes. These interactions can be also observed for the other two experiments including two GW hotspots. The combination of the RM and HI (EA and HI) GW hotspots shows that the impact of the RM (EA) GW hotspot on the polar vortex stability, normally leading to a strong deceleration of the zonal mean flow, is strongly dampened by the HI GW hotspot.

The combination of all three GW hotspots (Figure 3b—yellow contour line) shows that the negative zonal wind differences are enhanced compared to those of the RM + HI and EA + HI simulations. The same effect can be observed for the positive zonal wind differences at lower latitudes, being larger for the combination of all three GW hotspots. Thus, the RM and EA GW hotspots can partly maintain their influence in the polar region when they are both interfering with the HI GW hotspot. The deceleration (acceleration) of the zonal mean flow is about $-8\,\mathrm{ms}^{-1}$ ($8\,\mathrm{ms}^{-1}$) northward (southward) of $60°$ N. While in this run we have tripled the additional GW forcing, the zonal wind anomalies are not larger compared to the single and double GW hotspot simulations.

To summarize the effect interference of the doubled GW hotspots, there is a tendency for linear or constructive interference at lower latitudes mainly southward of $30°$ N, while the effects interfere destructively at middle to higher latitudes. The sum of the respective single GW hotspot, e.g., regarding zonal wind differences, is similar to the differences of the combined GW hotspots at lower latitudes, which means that the interaction of the effects is additive or rather linear. In contrast to the lower latitudes, the sum is different (more negative/positive) in comparison to the differences of the combined GW hotspots at middle and higher latitudes, which indicates nonlinear interference. The deviations between the summed up single GW hotspots and combined GW hotspots differences

at middle and higher latitudes are largest for the combination of the RM and HI GW hotspots. Thus, the effects of the RM and HI GW hotspots interact mostly destructive. These observed destructive (constructive) interferences between the GW hotspots at higher (lower) latitudes are caused by the meridional tilt of the local wind and temperature anomalies induced by the additional GWD forcing.

*3.2. Hotspot Effects on Distribution of the Geopotential Height and Vorticity*

The weakening of the polar vortex can be also seen in Figure 4 showing the geopotential height and potential vorticity differences northward of 30° N, which were averaged between 20 and 30 km. In general, the different combinations of all three GW hotspots lead to an increase (solid contour lines in intervals of 5 m) of the geopotential height at higher latitudes. These results correspond to the negative zonal wind anomalies in the polar region and also indicate a weakening of the polar vortex. In the region of the Aleutian high located between 150° E and 150° W (see Ref simulation in Figure 4h, we observe a negative geopotential height anomaly. Thus, the polar vortex is less disturbed by the Aleutian high and it can partly recover at lower latitudes, which is in accordance with the increasing zonal mean flow. This was also already reported by Samtleben et al. [16]. Besides the geopotential height differences we also calculated the potential vorticity differences, which are proportional to the vorticity and stratification changes and therefore, positive values illustrate the development of a cyclone. These differences are indicated by the color coding. The potential vorticity, which is normally increasing towards the polar region, shows a tendency for decreasing with latitude (mostly blue areas = negative anomalies) because of the displacement or weakening of the polar vortex.

Owing to the displacement of the polar vortex towards lower latitudes we observe positive potential vorticity anomalies at lower latitudes. These results are comparable to those of Kruse [57], who observed an anticyclonic potential vorticity polar cap and cyclonic potential vorticity banners at midlatitudes induced by mountain wave drag. Thus, our results clarify that our artificial GW enhancement acts like OGW drag in comprehensive models. According to the results of Samtleben et al. [17], the extrema of the positive (negative) potential vorticity anomalies are located at the southern (northern) flank of the respective GW hotspots whether in combination or as single GW hotspot. This may be explained by the quasi-geostrophic potential vorticity being inversely proportional to the negative zonal GWD, when neglecting diabatic processes and the meridional GWD. The negative potential vorticity anomalies are more pronounced at the northern flank of the HI or EA GW hotspots, when each of both GW hotspots are combined with the RM GW hotspot. However, the negative potential vorticity anomalies at the northern flank of the RM GW hotspot are less intense in combination with one of the other GW hotspots, although the single RM GW hotspot leads to the strongest decrease in the potential vorticity. Thus, it is an intensification of the effects of the HI or EA GW hotspot in combination with the RM GW hotspot, which normally has a larger impact (as single GW hotspot) on the circulation, as already observed in the zonal wind anomalies. In case of the HI + EA experiment the negative potential vorticity differences are enhanced (weakened) near the HI (EA) GW hotspot. The same effect can be seen in the geopotential height anomalies mainly intensified near the HI and EA GW hotspot. E.g., the RM GW hotspot (single: 65 m) shows less intense anomalies in combination (with EA: 50 m) with the EA GW hotspot compared to its caused anomalies as single GW hotspot.

Both combinations HI + RM and HI + EA including the HI GW hotspot Figure 4e, f also result in an intensification of the geopotential height anomalies (40 m) at the northern flank of the HI GW hotspot, while HI normally leads to the weakest anomalies as single GW hotpot compared to the other two GW hotspots Figure 4c. The combined RM + HI + EA simulation Figure 4h shows that despite the included HI GW hotspot the interactions of the RM and EA GW hotspots still lead to a strong increase in the geopotential height at the northern flank of the EA GW hotspot, which is not dampened by the HI GW hotspot. In this case the geopotential height anomalies are comparable to the EA and RM simulations in combination with enhanced positive geopotential anomalies at the northern flank of the HI GW hotspot.

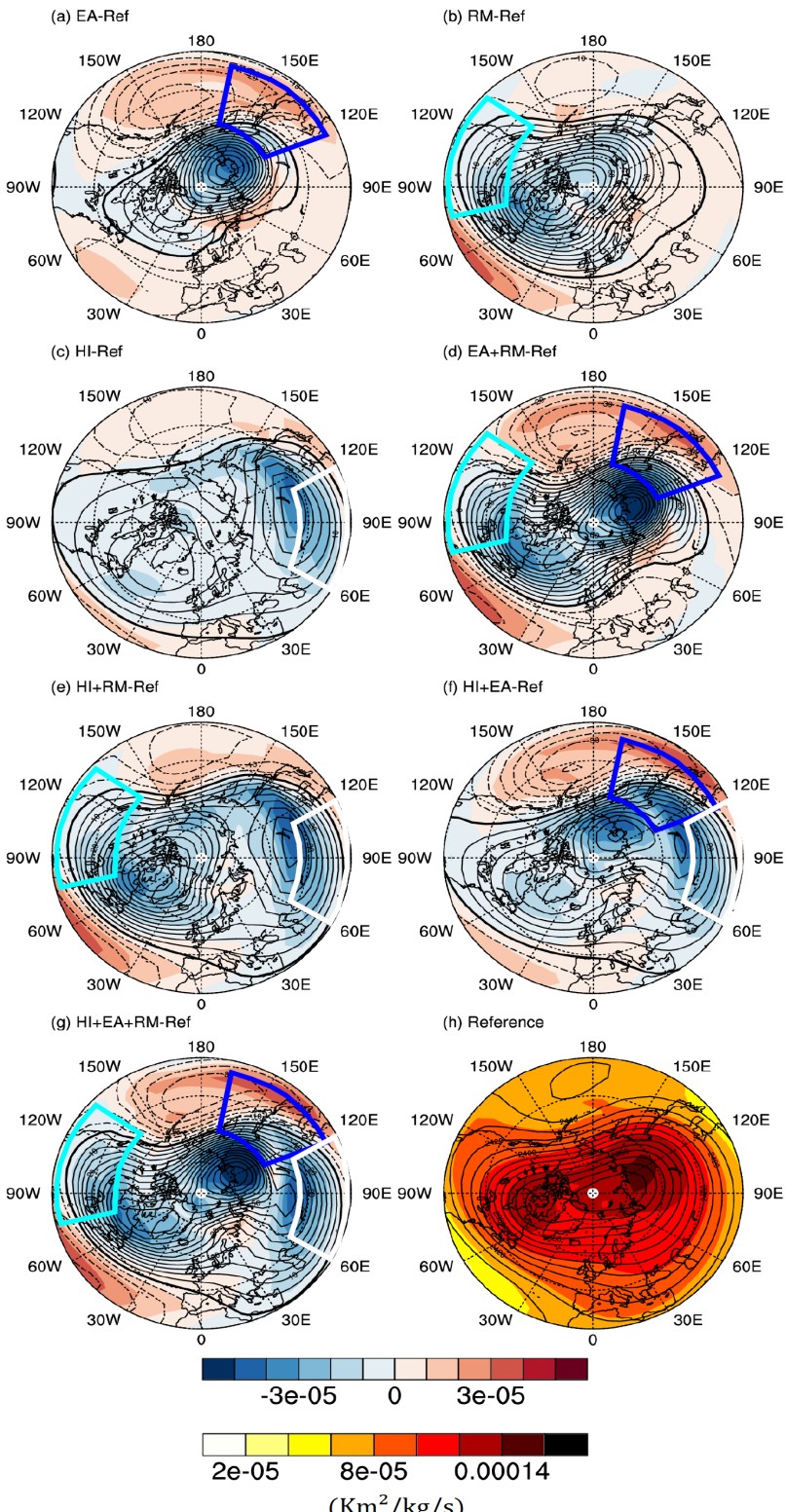

**Figure 4.** Geopotential height differences (contour lines) and potential vorticity differences (color coding) between each experiment (**a**)–(**g**) and the Ref simulation, averaged between 20 and 30 km. Latitudes range from $30°$ N to the pole. Geopotential differences are presented in intervals of 5 m. The zero lines are highlighted. Dashed (solid) lines represent negative (positive) differences. The positions of the GW hotspots are illustrated by the boxes. Geopotential heights and potential vorticity from the Ref simulation are shown in panel (**h**).



Also the potential vorticity anomalies are quite similar to those of the combined RM and EA GW hotspot so that they are not less pronounced owing to the HI GW hotspot. In the region of the HI GW hotspot the potential vorticity anomalies intensify as well as the increase of the geopotential height with values of more than $-45\,m$ at the northern flank of the HI GW hotspot. Thus, the combination of all three GW hotspots leads to more pronounced anomalies in the region of the HI GW hotspot compared to the single and 2-GW hotspots experiments.

In connection to the results in Figure 3 we have seen the weakening of the polar vortex at higher latitudes as well as its displacement towards lower latitudes (increase in the geopotential height and decrease in potential vorticity). As for the zonal wind and temperature anomalies, it can be seen that a combination of two GW hotspots always leads to an enhancement of the effects induced by the weakest (weak impact on circulation—e.g., small zonal wind anomalies, etc.) of both GW hotspots (EA in combination with RM, HI in combination with EA or RM, the weaker one is highlighted here). From the zonal means in Figure 3, we have not seen that the impact of the strongest (strong impact on circulation—e.g., large zonal wind anomalies, etc.) GW hotspot (RM in combination with HI or EA, EA in combination with HI) is not completely dampened by the weaker GW hotspot. In the region of the stronger GW hotspot the anomalies are still comparable to those of the referring single GW hotspot simulation despite the combination with a weaker GW hotspot. Thus, its influence is not completely reduced. This effect can be also observed in the tripled GW hotspot simulation.

## 4. Effect on Planetary Wave Activity – Amplitudes and Propagation Conditions

In this paragraph, we will present the effects of the GW hotspots, both single ones and in combination, on the development and propagation conditions of atmospheric waves. Therefore, we only capture the impact of the artificial GW forcing on the SPW activity but neglect the feedback mechanisms of the SPW changes on the GWD because those are small in comparison to the artificial GWD enhancement. We will concentrate on the SPWs in a specific altitude directly located above the artificial GW forcings by showing the SPWs amplitudes and Plumb fluxes. To see what is happening in the whole middle atmosphere we are mainly concentrating on the amplitudes and Eliassen–Palm (EP) fluxes of the SPW with wavenumber 1 (SPW 1), which is mainly affected by the GW hotspots.

### 4.1. Impact on the Atmospheric Wave Activity in the Region of the GW Hotspots

To get an overview of the included atmospheric waves during the different experiments, we performed a wavenumber-frequency analysis of the zonal wind of each experiment. This provides atmospheric waves with periods of 0hrs, which are the SPWs with wavenumbers from 1–3 and additional atmospheric waves with periods of 12 h (wavenumber 2) and 24 h (wavenumber 1). The latter are the migrating diurnal (24 h) and semidiurnal (12 h) tides. Because we are mainly focusing on the stratospheric effects, the tidal amplitudes, which are maximizing in the thermosphere, are neglected in our analysis. This is justified because their amplitudes are small in the stratosphere. This shows that the GW hotspots do not create different PWs than SPWs, which is an effect of the constant GW forcing not varying in time. This may be different when using an intermittent GW forcing. To present the latitudinal variability of the SPW amplitudes, Figure 5h shows latitude-wavenumber cross-sections of the SPW 1–3 zonal wind amplitudes in the Ref simulation for an altitude of 38 km, which is above the artificial GW forcing. The SPW 1 amplitude, which is ranging between $2.7\,ms^{-1}$ and $21.2\,ms^{-1}$ shows the largest amplitudes at lower to middle latitudes ($30°\,N–50°\,N$) and at higher latitudes (northward of $80°\,N$). In between the amplitudes minimize at the border of the polar vortex, where the west wind is strongest. Compared to the SPW 1 amplitudes, the SPW 2 and particularly SPW 3 amplitudes are smaller having values between $1.6\,ms^{-1}$ and $13.7\,ms^{-1}$. The latitudinal distribution of the maximum SPW 2 amplitudes is similar to the SPW 1 one, and they maximize between 40 and $50°\,N$ as well as around $75°\,N$. SPW 3 amplitudes vary between 0 and $5.4\,ms^{-1}$ and maximize at lower latitudes.

Panels (a–g) of Figure 5 show zonal wind amplitude differences between each GW hotspot experiment and the Ref simulation. In all cases, around 60° N the SPW 1 amplitude is mainly increasing in connection with a local GW forcing. The positive anomalies range between 0.9 ms$^{-1}$ HI GW hotspot—Figure 5c and 6.6 ms$^{-1}$ RM GW hotspot—Figure 5a. This represents an increase of more than 20 up to 150% compared to the Ref SPW 1 amplitude in this region. For most of the experiments except for the RM (Figure 5a) and EA simulation (Figure 5b) there is only one, partly very narrow, latitudinal range, where the SPW 1 amplitude is increasing. This already leads us to the first hypothesis that especially the interference of the GW hotspots as well as the HI GW hotspot may decrease the SPW 1 activity at midlatitudes, especially around 40° N.

In the RM GW hotspot simulation (Figure 5a) we observe an increase of the SPW 1 amplitude at the southern and northern flank of the GW hotspot as well as in the polar region. The same can be observed for the EA GW hotspot (Figure 5b), when neglecting the increase at its southern flank. This indicates that the intensified SPW 1 activity leads to larger energy and momentum transfer induced by breaking SPWs 1 for these two experiments. This may cause a deceleration of the zonal mean flow, which is strongest for the RM and EA simulations. In the region between 30° N–50° N, where we observed a maximum SPW 1 amplitude for the Ref simulation, the SPW 1 amplitude anomalies are strongly negative for all simulations.

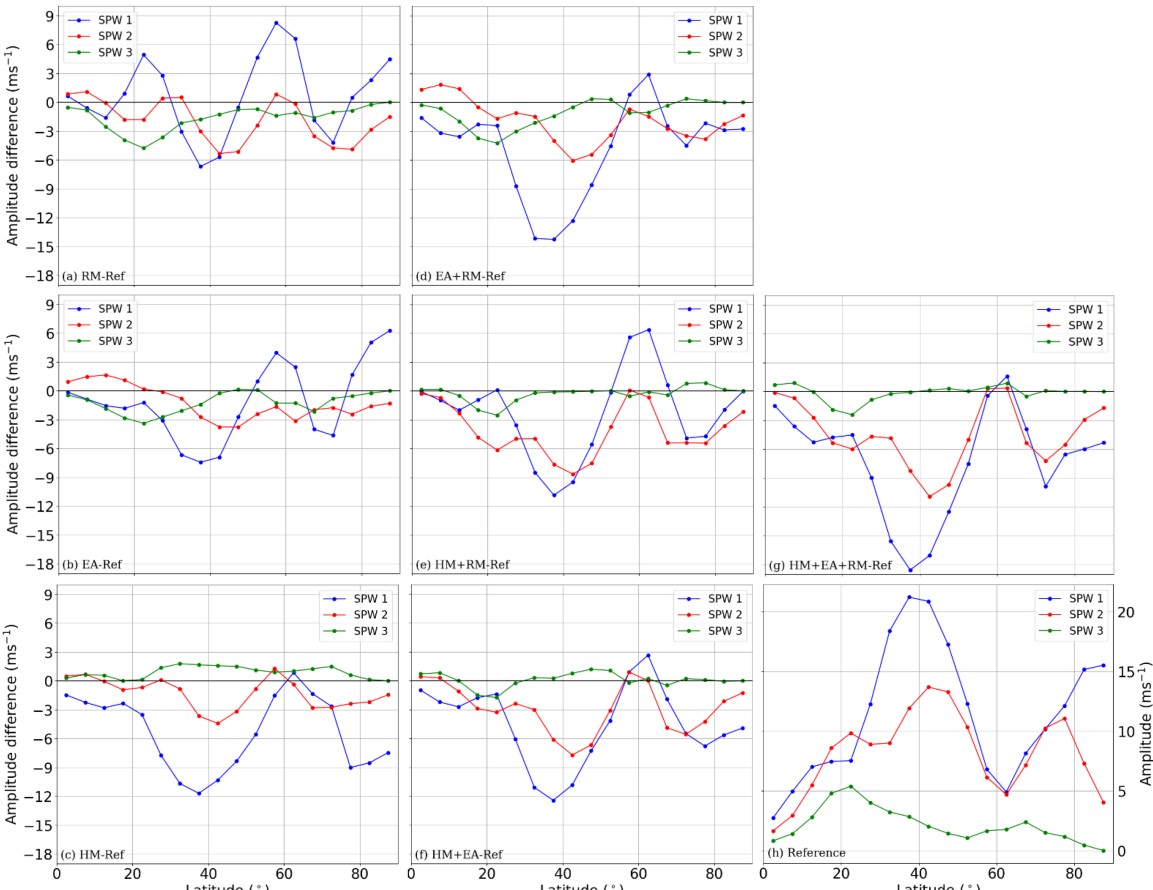

**Figure 5.** Latitude-wavenumber analysis of the zonal wind SPW from the reference simulation at 38 km (lower right panel), and the amplitude differences between each experiment and the reference simulation (h). The left column represents the differences for the RM (a), EA (b) and HI (c) simulations. The middle column represents those of the RM + EA (d), RM + HI (e) and HI + EA (f) simulations. The right column shows the differences for the RM + HI + EA (g) simulation.

This negative anomaly is strongest for the RM + HI + EA simulation (Figure 5g) with a decrease of $-18.6\,\mathrm{ms^{-1}}$. This shows that the propagation conditions significantly change owing to the combination of GW hotspots.

With respect to the SPW 2 amplitude, we mainly observe an increase of the SPW 2 activity at lower latitudes, where the SPW 1 activity is reduced. This is more pronounced for the EA GW hotspot in Figure 5b with SPW 2 maximum amplitude anomalies of $1.7\,\mathrm{ms^{-1}}$, and less pronounced for the RM in Figure 5a and HI GW hotspots in Figure 5c. While all three single GW hotspot simulations show the increase at lower latitudes, not all of the simulations including the interference of two GW hotspots correspond to these results. The combination of the RM and HI GW hotspots (HI + RM, Figure 5e) does not lead to larger SPW 2 amplitudes. Also, when the EA GW hotspot is added to these two GW hotspots (HI + EA + RM, Figure 5g the SPW 2 amplitude anomalies do not change at lower latitudes.

The SPW 3 amplitude anomalies are mainly negative for the RM and EA hotspots, but positive for the HI hotspot. For the RM and EA GW hotspot the SPW 3 activity is nearly completely dampened, while for the HI GW hotspot the SPW 3 amplitude has partly doubled. The interference of two GW hotspots partly lead to an increase of the SPW 3 amplitude, which is strongest for the HI + EA hotspot combination shown in Figure 5f. But the EA GW hotspot dampens the constructive effect of the HI GW hotspot on the SPW 3 activity, especially near the borders of the EA GW hotspot (blue lines). Thus, the SPW 3 amplitude anomalies are smaller compared to those caused by the single HI GW hotspot. These two GW hotspots in combination with the third RM GW hotspot cause again a weaker increase in some regions.

To investigate the wave behavior in detail, Figure 6 shows the three-dimensional wave activity flux (Plumb flux, [58]) at 38 km. It includes the wave activity flux of all SPWs. In Figure 6 the direction and the strength of the horizontal wave activity flux is illustrated by the colored arrows, while the vertical component is given by the edge color of the arrows. Blue (red) color indicates downward (upward) propagation. From the horizontal wave activity we observe one large area of strong wave activity, located above East Asia, Alaska and Canada, from where the horizontal wave activity flux is directed towards the tropical region. In the region of the maximized horizontal wave activity flux there is also an upward directed wave activity flux around $50^\circ$ N. Above the Atlantic and Europe the wave activity flux is less pronounced. SPW 1 is the dominating wave in our experiments, therefore, the Plumb flux is mainly determined by SPW 1, and the Plumb flux characteristics correspond to the properties of the SPW 1 EP flux reported by Samtleben et al. [16]. They analysed the same reference simulation and have shown that the SPW 1 is mainly propagating via the midlatitudes towards the tropical stratosphere.

Figure 7 shows the Plumb flux differences between each experiment including single and combined GW hotspots and the Ref simulation.

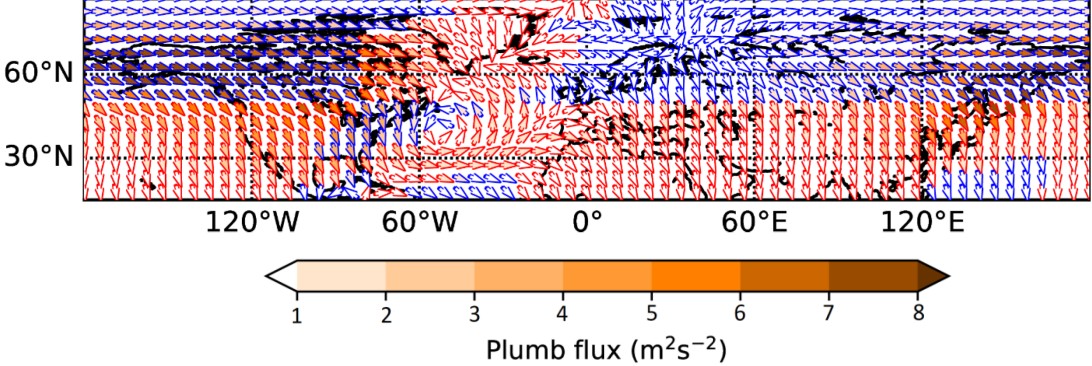

**Figure 6.** Three-dimensional Plumb flux for the Ref simulation at 38 km. The horizontal flux is illustrated by the colored arrows indicating the direction and strength. The vertical component is illustrated by the edge color of the arrows. Blue (red) indicates downward (upward) propagation.

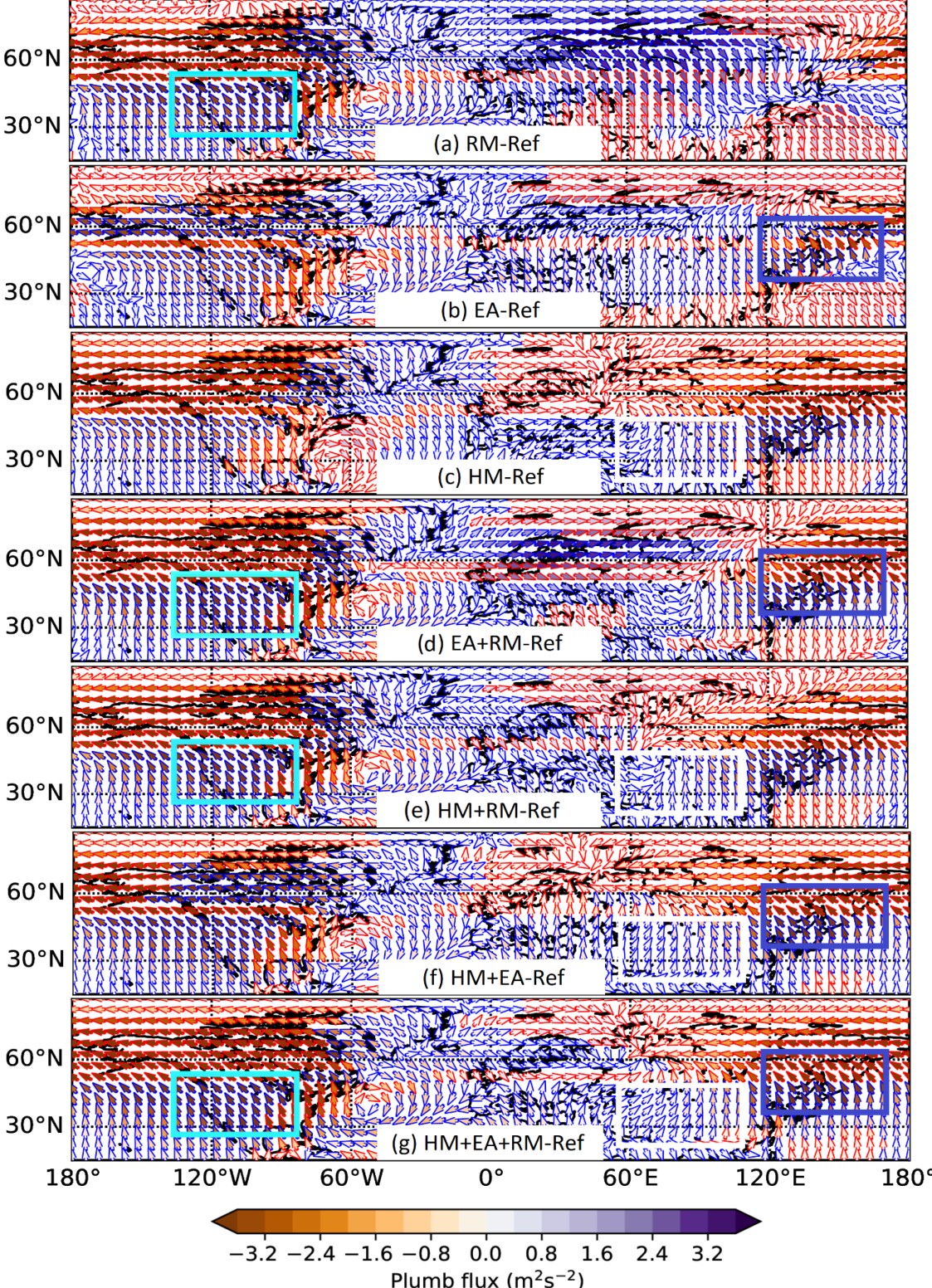

**Figure 7.** Three-dimensional Plumb flux differences between each simulation and the Ref simulation at 38 km altitude. The difference in the horizontal flux is illustrated by the colored arrows indicating direction and strength. The difference in the vertical component is illustrated by the edge color of the arrows. The position of the GW hotspots is illustrated by the boxes. RM − Ref (**a**), EA − Ref (**b**), HI − Ref (**c**), RM + EA − Ref (**d**), RM + HI − Ref (**e**), HI + EA − Ref (**f**), RM + HI + EA − Ref (**g**).

Blue (red) edged arrows indicate negative (positive) Plumb flux differences of the vertical component. In general, the horizontal wave activity flux is strongly decreasing above North America and the West Pacific region, which means that the single as well as the combined GW hotspots are acting as a block preventing wave propagation, i.e., energy transport. In this region the anomalies of the vertical component are also negative around 50° N. For the RM run in Figure 7a the apparent absence of the enhanced Plumb flux above North America is partly compensated by an increase above Eurasia, which, however, is mainly limited to higher latitudes. The vertical component of the wave activity flux over the Eurasian region around 60° N turns positive as well. Compared to the RM GW hotspot, the EA GW hotspot is not directly located in the region of maximum wave activity flux (in the Ref simulation) but on the left-hand side. This GW hotspot also reduces the wave activity flux above North America but not as strong as the RM GW hotspot does (Figure 7b). Also the increase of the horizontal wave activity flux over Eurasia is visible, but it is weak and is not connected with an increase in the vertical component.

While the HI GW hotspot Figure in 7c is not located near the maximum wave activity flux above North America in the Ref simulation, the wave activity is strongly decreasing. Especially, the horizontal wave activity density is strongly decreasing above East Asia as well as the vertical component around 40° N (less positive) and 50° N (less negative). Thus, the vertical transport of energy nearly vanishes and the wave activity has significantly weakened (see also Figure 5c). This GW hotspot also shows more often an SPW amplitude decrease (Figure 5) than the other two GW hotspots.

The RM + EA hotspot combination presented in Figure 7d leads to a strong Plumb flux decrease, which is even larger than for the single RM and EA GW hotspots. The horizontal wave activity flux anomalies are not only negative above North America but also in the West Pacific region, which is not the case in any of the single GW hotspot simulations. The positive horizontal wave activity flux anomalies above Eurasia are now more located above Scandinavia and are less (more) pronounced than in the RM (EA) single GW hotspot simulation. In this region there is also an increase in the vertical component of the wave activity flux, in contrast to any other simulations. Panels (e–g) show the HI + RM, HI + EA, and HI + EA + RM hotspot combinations. They all show similar patterns with a wave activity flux decrease above North America and East Asia, while the anomalies are strongest for HI + EA + RM.

To briefly summarize, the combined GW hotspots mainly lead to a decrease of the wave activity in connection with decreasing (i) amplitudes, especially at midlatitudes and (ii) fluxes above East Asia and North America, which means that the flux towards lower latitudes as well as the upward transport of energy is strongly dampened. This effect is partly compensated by the single RM and EA GW hotspot as well both GW hotspots in combination leading to an enhanced Plumb flux above Eurasia. Thus, the impact of the atmospheric waves, SPWs 1–3, is reduced and partly blocked by the respective GW hotspots and their interactions. As for the constructive and destructive interference at lower and higher latitudes this effect is caused by the meridional tilt of the SPW phases induced by the additional GWD forcing and the climatological eddies.

### 4.2. Impact on the SPW 1 Activity in the Middle Atmosphere

Because the SPW 1 shows the strongest amplitudes, we analyse here zonal wind SPW 1 amplitudes and the differences between each experiment and the Ref simulation as well as EP flux and EP flux divergence differences. Samtleben et al. [17] already analysed the SPW 1 anomalies induced by localized GW hotspots in detail. Owing to the broad consistency of some single GW hotspots to our experiments we briefly summarize the effects of the single RM, EA and HI GW hotspots. The contour lines in Figure 8 represent the SPW 1 amplitude distribution of the RM, EA, and HI experiment, resp., while the color shading illustrates the differences with the Ref simulation. For the RM hotspot in Figure 8a the SPW 1 anomalies are changing between positive and negative values, which are arranged in bands slightly tilted towards the polar region with increasing height. These results correspond to those in Figure 5a at 38 km.

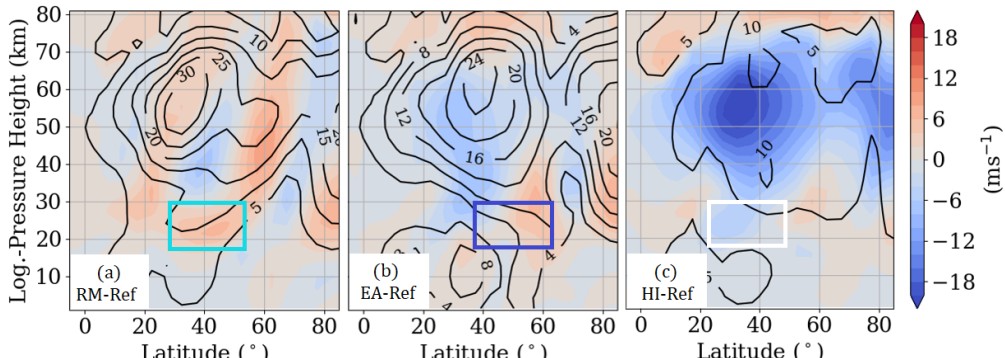

**Figure 8.** Stationary planetary wave 1 amplitude differences between the RM (**a**), HI (**b**), EA (**c**) experiments and the Ref simulation in color coding. The contour lines show the SPW 1 amplitude distribution of the respective GW hotspot. The color coding indicates the differences and the contour lines show the SPW 1 amplitude of the respective simulation.

As already reported by Samtleben et al. [16], the EA hotspot Figure 8b leads to a decrease of the SPW 1 amplitude at lower latitudes southward of 40° N and around 70° N. The positive SPW 1 amplitude anomalies of about 8 ms$^{-1}$ see also Figure 5b are restricted in altitude and only occur around 30 km at 60 and 80° N.

This positive anomaly may be induced by horizontally propagating SPWs 1 seen in the Plumb flux in Figure 7b . Overall the EA GW hotspot mainly leads to a SPW1 amplitude decrease [16]. The HI GW hotspot in Figure 8c shows strongly negative SPW 1 amplitude anomalies in the middle atmosphere turning positive only above 70 km and partly below 30 km. Consequently the SPW 1 amplitudes are very small and vary mostly between 0 and 5 ms$^{-1}$, only in few altitude ranges the SPW 1 amplitude exceeding 10 ms$^{-1}$. According to Samtleben et al. [17], the SPW 1 activity is mostly dampened because (i) the localized GW forcings prevent wave propagation at midlatitudes and (ii) the GW forcing interacts destructively with the original SPW 1, which is highly depending on the position of the GW hotspot with respect to the original SPW 1 phase. All three GW hotspots dampen the propagation at midlatitudes but the RM GW hotspot (similar to the H5 GW hotspot in Samtleben et al. [17]), which can be interpreted as an additional wave 1, interferes constructively with the original SPW 1 leading to increased SPW 1 amplitudes [17].

To investigate in how far the arrangement of the GW hotspots influences the interference of the respective GW hotspot effects, we now analyze the effect of the combined GW hotspots, EA + RM, HI + RM, HI + EA, and HI + EA + RM, and compare this with the linearly summed effect of each GW hotspot alone. Figure 9a–d shows the SPW 1 amplitude differences between each experiment and the Ref simulation in color coding. In addition, the contour lines represent the sum of the differences between the respective single GW hotspots and the Ref simulation (e.g., (RM − Ref) + (EA − Ref) in Figure 9a). Thus, differences between contours and shading indicate destructive or constructive interference between the combined GW hotspots, i.e., deviations from a linear superposition of the GW hotspot effects. The right column (e–h) illustrates the difference between the combined GW hotspot differences (color coding) and the added up differences (contour lines), e.g., (RM + EA − Ref) − ((RM − Ref) + (EA − Ref)). Dark red (blue) areas indicate a stronger increase (decrease) of the SPW 1 amplitude by the combination of the GW hotspots and therefore, a constructive interference of the combined GW hotspots with respect to the SPW 1 activity. Red (blue) indicates a less stronger decrease (increase) of the SPW 1 amplitude by the combined GW hotspots compared to the added up differences. Light red (blue) indicates a change in sign from negative (positive) anomalies for the added up differences to positive (negative) anomalies for the combined GW hotspots differences. To summarize, regions colored in blue show a stronger decrease of the SPW 1 amplitude induced by the combination of the GW hotspots, which means that there is a stronger destructive effect on the SPW amplitude induced by the combined GW hotspots than expected from linear interference.

We also introduce a linearity factor defined as the ratio of the effect of combined hotspots and the sum of single hotspot effects, e.g., for the combination of the RM and EA hotspots:

$$L = \frac{(RM + EA) - Ref}{(RM - Ref) + (EA - Ref)} \qquad (1)$$

If the differences (RM + EA) − Ref and (RM − Ref) + (EA − Ref)) are both either positive or negative, L (i) is positive and (ii) indicates an additive or rather linear interference when it is equal or larger than 1. In some cases both differences have an opposite sign, i.e., L (i) is negative and (ii) indicates a subtractive or destructive interference because the combined GW hotspots effect differs from the linear sum resulting from the respective single GW hotspots. In connection to the color coding, this means that dark red/blue regions in Figure 9 stand for constructive interference between the GW hotspots, while the other colored gradations represent destructive interference.

Starting with the SPW 1 amplitude differences between each experiment and the Ref simulation (Figure 9a–d color coding), it can be generally seen that the SPW 1 amplitude is strongly decreasing between 30 and 70 km, with the largest negative anomalies around 40 and 70° N. The effect is strongest for the experiment including all three GW hotspots. In parts of the lower stratosphere, southward of 20° N and northward of 60° N as well as above 70 km, the SPW 1 amplitude is mostly increasing. The enhanced amplitudes around 60° N, which is largest for the RM and EA combination, may be caused by horizontally propagating SPW 1 according to the increase of the horizontal component of the Plumb flux, which we observed in case of the EA and RM GW hotspots. The summed up single GW hotspot differences and the combined GW hotspot differences mutually vary and need to be discussed in more detail.

By comparing the values of the combined EA and RM GW hotspots differences to the added up differences, it can be seen that the negative SPW 1 amplitude changes at midlatitudes and in the polar region above 30 km are more pronounced (more negative = dark blue) for the combined GW hotspots simulation (Figure 9e). Thus, with respect to the SPW 1 activity we observe a destructive effect caused by the combined GW hotspots. According to the linear superposition of the single GW hotspots the combined GW hotspots lead to similar differences or even exceed those, i.e., we observe an additive or linear interference between both GW hotspots in these regions colored in dark blue also indicated by the linearity factor being strongly positive. Around 60° N, where the added up differences of the single EA and RM GW hotspots are strongly positive above 30 km (Figure 9a, contour lines), the SPW 1 amplitude differences in Figure 9e show that the SPW 1 amplitude changes are less positive (blue) or even negative (light blue) for the combined GW hotspots in this region. Thus, above 30 km we mainly observe an intensified decreasing SPW 1 amplitude owing to the combination of the EA and RM GW hotspots. In this case, both GW hotspots interfere destructively emphasized by the negative and slightly positive (<1) L in this region. Solely, the positive SPW 1 anomalies at the northern flank of the EA GW hotspot seen in (Figure 9a, red regions) are more positive as they would be if we sum up the differences of the single GW hotspots. In this region the linear interference of both GW hotspots (linearity factor larger than 1) lead to a constructive effect on the SPW 1 amplitude. Because of the negative SPW forcing originating from the combination of the RM and EA GW hotspots, most of the regions (77% of the presented latitude-height plot) are colored in blue in Figure 9e.

The SPW 1 amplitude changes of the combined HI and RM GW hotspots correspond well to the summed up differences of the single HI and RM GW hotspots. The difference between these two differences in Figure 9f shows that the SPW 1 amplitude differences caused by the combined GW hotspots are less negative than taken from linear addition in most of the regions between 30 and 70 km (colored in red). Thus, the combination of both GW hotspots has a less destructive effect on the SPW 1 activity than expected from linear assumption. In the red region the linearity factor ranges between 0 and 1 indicating destructive interference between both GW hotspots because the combined GW hotspots difference is less negative.

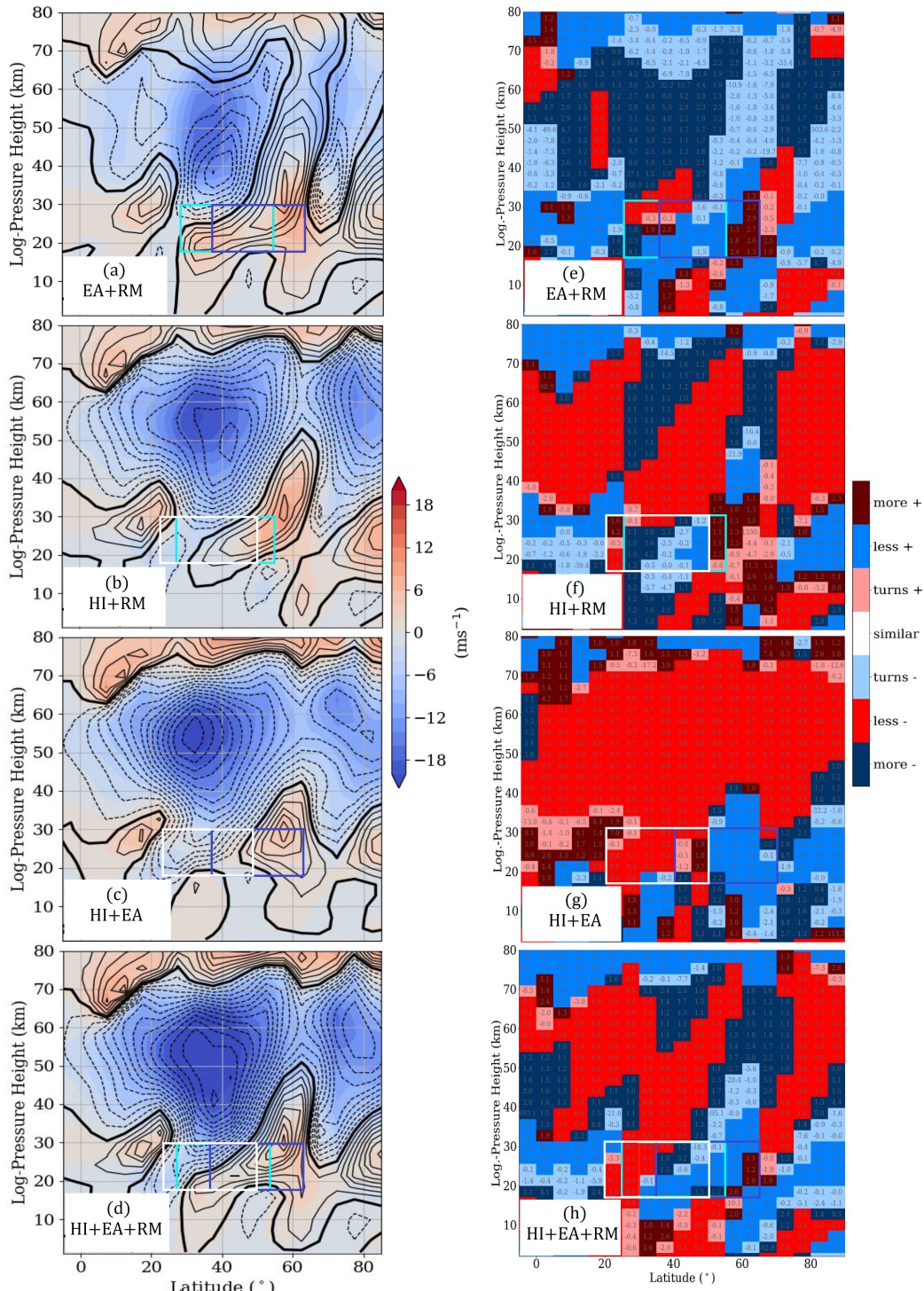

**Figure 9.** Stationary planetary wave 1 amplitude differences between the RM + EA (**a**), RM + HI (**b**), HI + EA (**c**) and RM + HI + EA (**d**) GW hotspots experiments and the Ref simulation in color coding. The contour lines show the sum of the single GW hotspot differences: (RM − Ref) + (EA − Ref) (**a**), (RM − Ref) + (HI − Ref) (**b**), (HI − Ref) + (EA − Ref) (**c**) and (RM − Ref) + (HI − Ref) + (EA − Ref) (**d**). The right figures show the differences between the two differences shown in the left figures: (RM + EA − Ref) − ((RM − Ref) + (EA − Ref)) (**e**), (RM + HI − Ref) − ((RM − Ref) + (HI − Ref)) (**f**), (EA + HI − Ref) − ((EA − Ref) + (HI − Ref)) (**g**) and (RM + HI + EA − Ref) − ((RM − Ref) + (HI − Ref) + (EA − Ref)) (**h**). Blue (red) areas indicate destructive (constructive) interference with respect to the stationary planetary wave (SPW) 1 activity. The numbers represent the linearity factor L (Equation (1)).

There are only a few regions, in which the negative SPW 1 amplitude anomalies are more pronounced and both GW hotspots interfere constructively. The positive SPW 1 amplitude changes around 60° N caused by the combined GW hotspots are slightly enhanced compared to those obtained from the linear sum of the single GW hotspots, while the SPW 1 amplitude changes of the combined GW hotspots in the polar region are less intense. Comparing the SPW 1 changes induced by the combined RM and HI GW hotspots in Figure 9b with the single RM and HI GW hotspots, it can be seen that the impact on the SPW 1 activity is less destructive. The negative SPW 1 amplitude changes caused by the combined HI and RM GW hotspots are less intense than those of the summed up differences of the single GW hotspots in connection with linearity factors smaller than 1 standing for destructive interference. This means that the combination of these two GW hotspots leads to a weaker decrease of the SPW 1 activity compared to the sum of the respective single GW hotspots. Only 45% of the area in Figure 9f is blue (destructive effect on SPW 1). The results of the experiment including all three hotspots are similar to those of the combined RM and HI GW hotspots. Solely around 60° N, the positive SPW 1 anomalies induced by the combination of all three hotspots is not as strong as the sum of the anomalies based on the respective single GW hotspots. Thus, the addition of the EA GW hotspots leads to a destructive interference between all three GW hotspots in this region.

Compared to all the other experiments, the combined EA and HI GW hotspots shown in Figure 9f lead to a large extent to smaller negative (positive) anomalies than expected from the linear assumption of the respective single EA and HI GW hotspots. Thus, the combined GW hotspots (i) have a less destructive (constructive) effect on the SPW 1 activity and (ii) their interactions are nonlinear or rather non-additive underlined by the positive L < 1. The less negative SPW anomalies caused by the combined GW hotspots predominate. The effect on the SPW 1 activity is less destructive for the EA and HI combination than for the other experiments. Therefore, this combinations also has the largest percentage of nonlinear interference between these two GW hotspots. In this experiment only 31% of the area in Figure 9g is blue (destructive interference). Up to 30 km northward of 60° N the SPW 1 differences caused by the combined GW hotspots are more negative or less positive. Thus, the combination has a destructive effect on the SPW 1 amplitude in this region.

To summarize, the combinations of the different GW hotspots mainly lead to a decrease of the SPW 1 activity. The combined EA and RM GW hotspots have a more destructive effect on the SPW 1 activity because of their additive interference. The SPW 1 amplitude differences of the combined EA and RM GW hotspots are more negative than those of the summed up SPW 1 amplitude differences based on the single GW hotspots. Compared to the EA and RM GW hotspots combination, the combined RM and HI GW hotspots have a less destructive effect on the SPW 1 activity owing to the nonlinear interference. In longitude, the RM and HI are displaced by 170° against each other; this is the reason, why a nearly complete counteracting of these two GW hotspots might be expected. The longitudinal distance between the EA and RM GW hotspots is only 105°. Furthermore, the EA GW hotspot is placed 10° further North, while the HI GW hotspots is located 5° further South than the RM GW hotspot. This spatial distribution explains, why the interference between the EA and RM GW hotspots is more constructive than the one between the RM and HI GW hotspots. The HI and the EA GW hotspots are located next to each other, however, this combination has the most constructive effect on the SPW 1 activity owing to the nonlinear interference. Thus, the zonal displacement between both GW hotspots of 20° leads to the non-additive interference. Another factor is the interference of the GW forcing with the original SPW 1. Samtleben et al. [17] showed that that the RM is mainly in phase with the original SPW 1, while the HI GW hotspot is mostly out of phase. We observe a weakening (strengthening) of the GW hotspot effects, which is more in (out) phase with the original SPW 1, when it is combined with a GW hotspot being more out (in) of phase with the original SPW 1. This also indicates destructive interference between the GW hotspots.

According to Samtleben et al. [17], single GW hotspots lead to a blocking of SPW 1 propagation at midlatitudes. Figure 10 shows the differences of the EP flux and its divergence between the respective experiment and the Ref run, in order to see if this is also the case for the combined GW hotspots.

Changes in the direction and strength of the EP flux difference is illustrated by the colored arrows, while the changes of the EP divergence are given in color coding. Areas of negative EP divergence in the Ref simulation are hatched to be able to interpret the EP divergence differences. From observations and also previous publications we know that there is one major branch of SPW 1 propagation originating at midlatitudes and from there going further upward and heading towards the equatorial stratosphere. And a second branch heading from the midlatitudes towards the polar lower stratosphere is less well-developed [16].

In all cases, the EP flux and, for large areas also its divergence, is decreasing as a result of the hotspot effect. An exception is the RM hotspot (Figure 10a), where the EP flux is increasing, especially at higher latitudes up to 60 km. In this region SPWs do normally not propagate owing to the strong polar vortex. Because of its displacement or even weakening, which is strongest for the RM GW hotspot (strong decrease in zonal mean flow and geopotential height), in this simulation the SPWs 1 are also propagating via the polar region into the middle atmosphere.

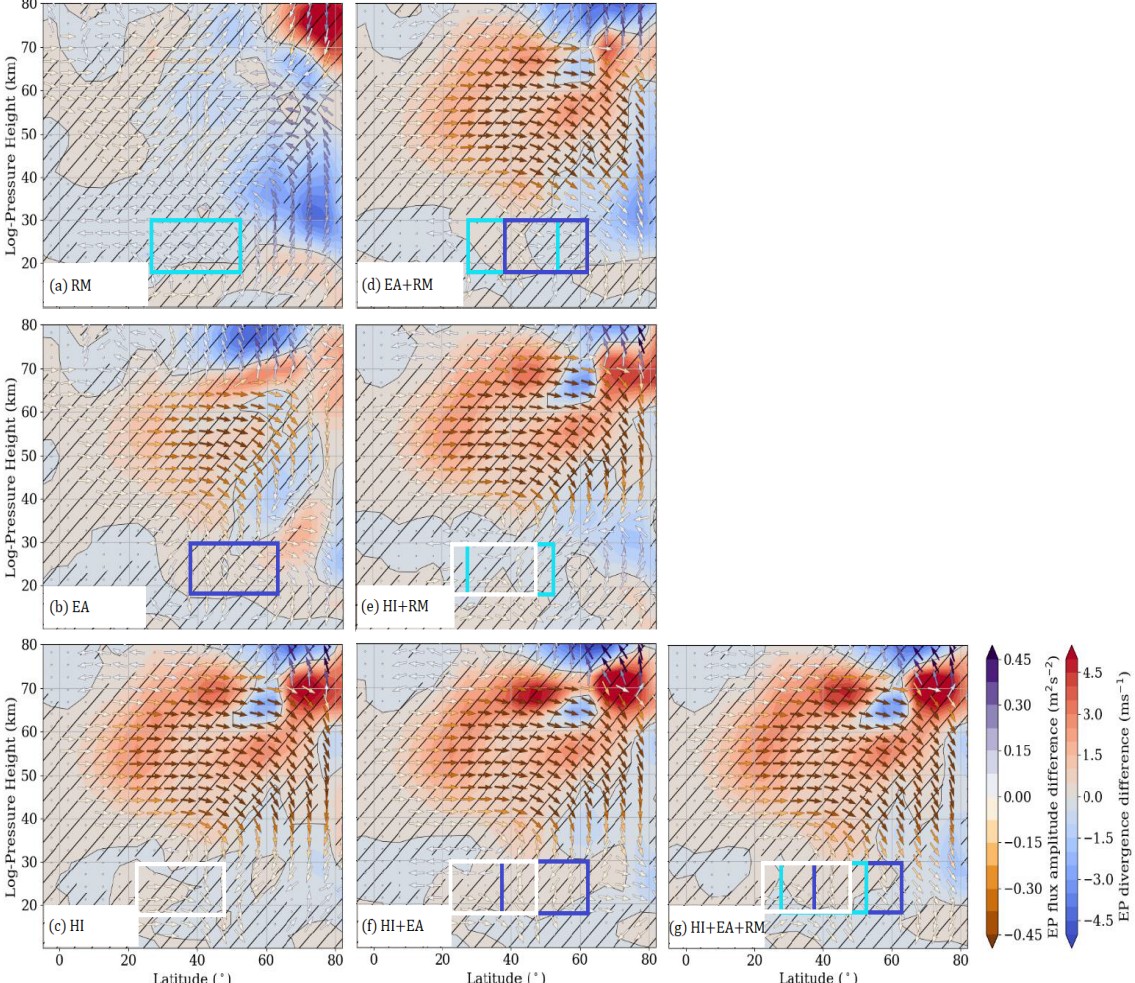

**Figure 10.** Colored shaded areas: EP flux and EP flux divergence differences of the SPW 1 between each experiment and the Ref simulation. The left column shows the RM (**a**), EA (**b**) and HI (**c**) GW hotspots. The middle column shows the RM + EA (**d**), RM + HI (**e**) and HI + EA (**f**) GW hotspots. The right column shows the RM + HI + EA GW hotspots (**g**). The color coding indicates the EP divergence differences; the zero line is highlighted. Areas of negative EP divergence in the Ref simulation are hatched. The arrows represent the changes in direction and strength of the EP flux between the respective experiment and the Ref simulation. The arrows were replaced by dots when the EP amplitude difference was smaller than 1% of the maximum EP flux amplitude difference.

Also more SPWs 1 are breaking in the polar region indicated by the negative EP divergence, which means that more energy and momentum is transferred leading to the deceleration of the west wind, and thus, to the destabilization of the polar vortex. In the polar upper mesosphere there is a strong positive EP divergence anomaly and therefore, represents a source of SPWs 1. But because of the wind reversal in the mesosphere, these SPWs 1 are not able to propagate further upward.

According to Samtleben et al. [17] the EA GW hotspot (Figure 10b) shows a decrease in the EP flux amplitude and the difference arrows are pointing downward. Thus, also the EP divergence is less negative or even positive, which explains the positive EP divergence anomalies, especially in the middle atmosphere at lower latitudes. This decrease of energy transfer normally induced by breaking SPWs leads to the acceleration of the zonal mean flow in this region observed in Figure 3b. In the upper mesosphere the EP flux is slightly increasing originating from a region of positive EP divergence (60° N, 70 km), a sink of SPWs 1. These SPWs 1 are mainly generated by local instabilities [16]. They propagate upward, but they break nearly immediately and lead to the negative EP divergence in the mesosphere. While the vertical propagation conditions changed dramatically, we observed positive SPW 1 amplitude anomalies around 30 km at midlatitudes and in the polar region lower stratosphere in Figure 8b. These positive anomalies are probably induced by horizontally propagating SPWs 1 observed in the increased Plumb flux above Eurasia (Figure 7b), which are not able to propagate further upward and therefore, are confined to specific altitude ranges. The HI GW hotspot effects in Figure 10c show a similar structure as the one for the EA GW hotspot. Also, the combinations of several GW hotspots show nearly the same behaviour in the EP flux and its divergence like the HM GW hotspot.

Thus, we may briefly summarize that the analysed GW hotspots both as single ones or in combination mainly lead to a dampening of the SPW 1 activity by preventing them from propagating upwards at midlatitudes. Thus, the GW hotspots act like a blockade. Owing to the suppressed propagation of SPWs 1 we observe a decrease in the EP flux in connection with a decreasing transfer of momentum and energy indicated by positive EP flux divergence anomalies. As an exception, in case of the RM GW hotspot the SPWs 1 are able to propagate via the higher latitudes into the middle atmosphere and compensate the damping. This is possible because of the stronger weakening of the polar vortex in the RM experiment, compared to the other experiments.

## 5. Conclusions

In this study, in addition to studies on the effects of single local GW hotspots, we performed new experiments concentrating on the interference between two or three different GW hotspots. These three GW hotspots were chosen guided by literature, and they are partly located in the same latitudinal belt: in the East Asian region (EA) [26], near the Himalayas (HI) [59] and at the Rocky Mountains (RM) [60]. Thus, most of these GW hotspots are mainly generated by orography as well as by convection and seismic activity [23,24]. To be able to interpret the interference of the GW hotspots we performed experiments including only one GW hotspot (3 experiments), two GW hotspots (3 experiments) and all three GW hotspots (1 experiment).

With respect to the results for the combined GW hotspots we directly see that the strongest decrease, e.g., in the zonal mean flow (RM GW hotspot) is not exceeded by the combined GW hotspots. Also the experiment including all three GW hotspots shows changes, which are less pronounced than those of the single RM and EA GW hotspots. However, at lower latitudes, e.g., the positive zonal wind anomalies are often the sum of the respective single GW hotspots zonal wind anomalies. Therefore, we observe an additive (nonlinear) interference of the combined GW hotspots at lower (higher) latitudes. Thus, the GW hotspot being out of phase partly dampens the effects of the GW hotspot being in phase. Another reason on the basis of the chosen GW hotspots is the position of the GW hotspots relative to each other. The RM GW hotspot is displaced by 170° (105°) related to the HI (EA) GW hotspot. But also the, e.g., negative zonal wind anomalies of the HI + EA GW hotspots simulation are less pronounced than in the EA GW hotspot simulation although these GW hotspots

are next to each other. Thus, also the latitudinal displacement needs to be considered, which is about $20°$ between the EA and HI GW hotspots (largest in our experiments). As a consequence an additional study including (i) other GW hotspots (close to the RM GW hotspot), which are also in phase with the modeled SPW in combination with the RM GW hotspot or (ii) other GW hotspots having the same latitude may substantiate the assumptions regarding the latitudinal and longitudinal displacement of the GW hotspots relative to each other.

With respect to the SPW 1 activity the increased SPW 1 amplitudes observed for the RM GW hotspot are suppressed by the HI GW hotspot as well as by the EA GW hotspot, or even by both together. Only the increase of the SPW 1 amplitude at the northern flank of the RM GW hotspot remains. The dampening of the SPW activity in the middle atmosphere is also illustrated by the Plumb flux including all SPWs, and the SPW 1 EP flux showing a strong decrease in the horizontal and vertical wave fluxes. Owing to the combination of the GW hotspots the wave activity comes nearly to a standstill. The enhanced SPW 1 EP flux of the RM GW hotspot via the polar region into the middle atmosphere is completely reduced by the other two GW hotspots preventing SPW propagation. Only the combination of the RM GW hotspot with the EA GW hotspot still leads to an enhanced horizontal wave flux above Eurasia but both single GW hotspots also exhibit this characteristic. As for the comparison of the summed up zonal wind differences of the single GW hotspots and the zonal wind combined GW hotspots, we mostly observe destructive interference between the combined GW hotspots. In most of our experiments the decrease as well as the increase of the SPW 1 amplitudes is less pronounced than expected from a purely linear effect based on the single GW hotspots. Nevertheless, the resulting changes are always weaker (stronger) than the anomalies caused by the single GW hotspot, if the latter is more in (more out) of phase with the modeled SPW.

This study provides first insights into the interference of realistic and mostly orographic GW hotspots on the winter northern hemisphere, showing that the combined GW hotspots mainly lead to a breakdown of the SPW activity. Because we only include three different GW hotspots, the results may look different if we would consider GW hotspots, which are also close to the RM GW hotspot (e.g., above the North American Great Plains according to Hoffmann et al. [61]). Nevertheless, this study is strongly simplified by choosing the same size, the same forcing and a continuous interference of these GW hotspots. To improve our approach we would need further analysis of observations including potential energy data or GWD (although these are partly biased [29,62,63]) data to determine the size of these GW hotspots. Because there are no compiled datasets of 3D daily observations there would be the option to use, e.g., 3D Canadian Middle atmosphere Model (CMAM) GWD data [64] to get a time series of the GW hotspot activity and also to see the strength of the GW forcing.

**Author Contributions:** Conceptualization, N.S.; Data curation, N.S.; Formal analysis, N.S. and C.J.; Investigation, N.S.; Methodology, N.S. and A.K.; Project administration, N.S. and C.J.; Software, N.S.; Supervision, C.J.; Validation, C.J.; Visualization, N.S., P.Š. and C.J.; Writing—original draft, N.S.; Writing—review and editing, A.K., P.Š., P.P. and C.J. All authors have read and agreed to the published version of the manuscript.

**Funding:** N.S., A.K., and C.J. gratefully acknowledge the funding by the Deutsche Forschungsgemeinschaft (DFG, German Research Foundation) through projects JA 836/32-1 and JA 836/43-1. P.S. is supported through the project *CZ.02.2.69/0.0/0.0/19_074/0016231* (International mobility of researchers at Charles University (MSCA-IF III)) for the research stay at BOKU Vienna, through the Czech Science Foundation (GA CR) grants nos. 16-01562J and 18-01625S and acknowledges discussions in the New Quantitative Constraints on OGW Stress and Drag team at the International Space Science Institute in Bern, Switzerland. We acknowledge support from Leipzig University for Open Access Publishing.

**Conflicts of Interest:** The authors declare no conflict of interest.

## Abbreviations

The following abbreviations are used in this manuscript:

| | |
|---|---|
| AO | Atlantic Oscillation |
| CMAM | Canadian Middle atmosphere Model |
| EA | East Asia |
| ENSO | El Nino Southern Oscillation |
| EP | Eliassen–Palm |
| EUV | Extreme ultra violet |
| GCM | Global circulation model |
| GPS | Global positioning system |
| GW | Gravity wave |
| GWD | Gravity wave drag |
| $GWD_u$ | zonal gravity wave drag |
| $GWD_v$ | meridional gravity wave drag |
| $GWD_T$ | heating due to breaking gravity waves |
| HI | Himalayas |
| L | Linearity factor |
| MUAM | Middle and upper atmosphere model |
| NAO | North Atlantic Oscillation |
| NH | Northern Hemisphere |
| OGW | Orographic gravity wave |
| PW | Planetary wave |
| QBO | Quasi-Biennial Oscillation |
| Ref | Reference |
| RM | Rocky Mountains |
| SH | Southern Hemisphere |
| SPW | Stationary planetary wave |
| SSW | Sudden stratospheric warming |

## Appendix A. Figures

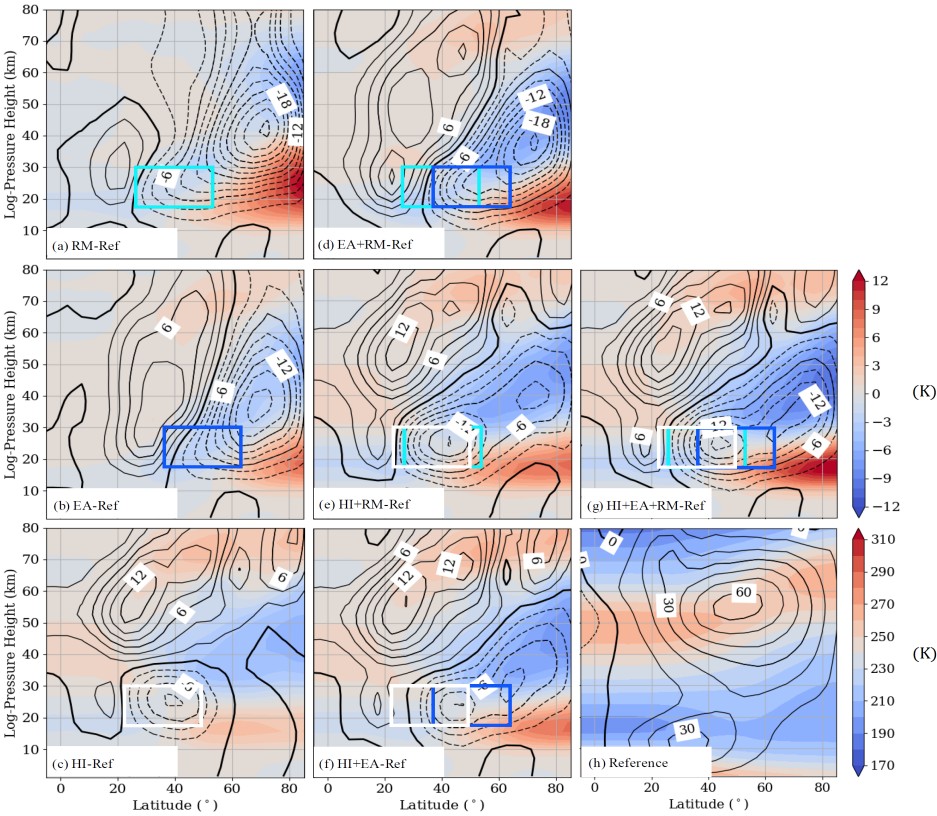

**Figure A1.** Zonal mean zonal wind differences (contour lines-in steps of $2\,\text{ms}^{-1}$) and temperature differences (colored) between the single GW hotspots (**a**)–(**c**), the doubled GW hotspots (**d**–**f**) and the tripled GW hotspots (**g**) and the Ref simulation. (**a**) RM (**b**) EA (**c**) HI (**d**) RM + EA (**e**) RM + HI (**f**) HI + EA (**g**) RM + HI + EA. Temperature (color coding) and zonal wind (contour lines) of the Ref simulation are added in the lower right panel (**h**). The upper color bar refers to the temperature differences while the lower color bar refers to the temperature of the Ref simulation.

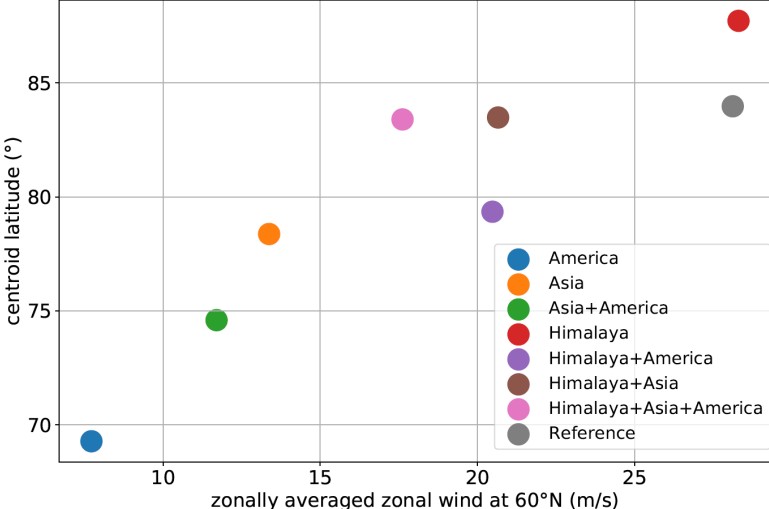

**Figure A2.**    Position of the polar vortex based on the centroid latitude (according to Matthewman et al. [65], Seviour et al. [66]) in relation to the changes in wind speed for each experiment at an altitude of 35 km.

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
