# Peer review of "Mutual Interference of Local Gravity Wave Forcings in the Stratosphere"

_atmosphere, doi:10.3390/atmos11111249_

Round 1
Reviewer 1 Report
In this study authors studied the effects of single local GW hotspots as well as the interference between two or three different GW hotspots. They reported an additive (nonlinear) interference of the combined GW hotspots at lower (higher) latitudes. This study provides first insights into the interference of realistic and mostly orographic GW hotspots on the winter northern hemisphere, showing that the combined GW hotspots mainly lead to a breakdown of the SPW activity.
I recommend this paper for publication in the present from. Just two comments:
Abbreviations like «SPW 1» in abstract have to be explained or not used.
Lines 137-139: " However, the mesospheric jet maximum southward of 60N and above 50 km, exceeding 60 ms−1 , is somewhat overestimated by about 10-20 ms−1 ."
May be jet maximum is overestimated in comparison to CIRA-86, but not in comparison to experiments. See, for example, https://core.ac.uk/download/pdf/190975353.pdf
Author Response
Dear Reviewer #1,
thank you for your comments. We attached a file including a point-by-point response to your comments.
Kind regards

Reviewer 2 Report
Review of the paper by Nadja Samtleben et al. “Mutual interference of local gravity wave forcings in the stratosphere”.
This paper numerically studies (using the Middle and Upper Atmosphere Model (MUAM) ) the combined impact of breaking orographic gravity waves (GWs) generated by the Rocky Mountains (RM), the Himalayas (HI) and the East Asian (EA) mountain region along with the planetary waves (PWs) on the global circulation of the middle atmosphere and ,particularly, on the polar vortex dynamics. From my point, there are several novel results obtained in the paper that deserve to be published in the journal Atmosphere:
- One of the important results is that an interference of GWs from RM, HI and EA is not linear and mostly has a destructive effect on propagation and generation of stationary planetary waves (SPW). Generally, the amplitudes of SPW, the Eliassen-Palm flux and Plumb flux are reduced due to interaction of GW contributions from RM, HI and EA.
- In general, the different combinations of contributions from all the three GW sources (RM, HI and EA) “ …lead to a warming of the polar stratosphere in combination with a decreasing zonal mean zonal wind at middle and higher latitudes. Thus, the additional GW forcings lead to a weakening of the polar vortex, which can be interpreted as a preconditioning of the polar vortex that may result in a SSW “(stratospheric warming).
However, there are still questions to be addressed in the paper before its publishing.
- The authors do not define in the paper the main parameters (among many others) in their numerical model that affect the interaction of GWs and SPW. How sensitive are the results obtained to the root mean square slopes of the mountain regions RM, HI and EA and their horizontal sizes?
- The parameterization of the gravity wave drag in global circulation models is still an open question. It is necessary to indicate what kind of parameterization scheme is used in the model and why.
- Does the model take into account the wave drag from propagating mountain waves with nonzero phase speed generated by varying in time wind, or only stationary mountain waves (with zero phase speed) caused by constant wind?

Author Response
Dear Reviewer #2,
thank you for your comments. We attached a file including a point-by-point response to your comments.
Kind regards

Reviewer 3 Report
This paper contains some new interesting developments concerning wave propagation and wave sources in Earth’s atmosphere. The authors have performed new simulations concentrating on the interference between two or three different gravity wave hotspots. Most of the considered gravity wave hotspots are mainly generated by orography. The authors have shown, in particular, that gravity wave hotspots interfere mostly nonlinearly. Furthermore, interfering gravity wave hotspots mostly have a destructive effect on stationary planetary wave propagation and generation. The results can be of interest for scientists studying Earth’s atmosphere, waves, nonlinear processes, etc.
The paper is written in a clear manner. However, in the present form, the paper cannot be published in Atmosphere. The reasons for this are the following.
- The authors consider gravity wave hotspots located in part at the same latitudinal belt: in the East Asian region, near the Himalayas and at the Rocky Mountains. However, these regions are subjected often to seismic processes, which can be a significant source of atmospheric wave activity. In particular, generation of internal gravity waves in the atmosphere above seismoactive regions is a typical phenomenon (see, e.g., V. V. Adushkin, V. I. Nifadiev, B. B. Chen, S. I. Popel, G. A. Kogai, A. Yu. Dubinskii, and P. G. Weidler, Variations of the Parameters of Internal Gravity Waves in the Atmosphere of Central Asia before Earthquakes, Doklady Earth Sciences 487, Part 1 (2019), 841-845; V. V. Adushkin, V. I. Nifadiev, B. B. Chen, S. I. Popel, G. A. Kogai, A. Yu. Dubinskii, and P. G. Weidler, Characteristics of Internal Gravity Waves and Earthquake Prediction, Doklady Earth Sciences 493, Part 2 (2020), 632-635). The authors have to include some discussion concerning this item.
- The paper contains some abbreviations which are not spelled out (e.g., in Abstract the abbreviation “SPW” is not spelled out). The authors have to check and to correct the manuscript.
If the authors reflect the above items adequately, then the publication of the paper can be possible.
I would like to review a revised version of the paper.
Author Response
Dear Reviewer #3,
thank you for your comments. We attached a file including a point-by-point response to your comments.
Kind regards

Reviewer 4 Report
Review of “Mutual interference of local gravity wave forcings in the stratosphere” by Samtleben et al.
This paper investigates how the combined forcings from three gravity wave (GW) hotspots in the Rocky Mountains, the Himalayas, and the East Asian region affect the northern winter stratosphere. In general, the interference between GW hotspots acts to dampen planetary wave activities in the stratosphere, leading to weaker stratospheric circulation changes compared to single Rocky Mountain GW hotspot forcing.
This paper is not easy to read. The results are not fully explained. The conclusions are not clear. Overall, the authors should put more efforts in interpreting the results, not just documenting the differences from the combined GW hotspots. The manuscript needs a major revision before it can be published.
Section 3.1: Why the Rocky Mountain GW hotspot has the largest impact on high latitude temperature and zonal wind? Why the GW interference is destructive in high latitudes and constructive in low latitudes?
Figure 5: It might be easier to just show a plot of wave amplitude as a function of latitude for each wavenumber.
Figures 6 and 7: These figures are confusing with similar color schemes for different variables. Consider using the length of the arrows to represent the magnitude of the horizontal flux.
Lines 417-423: The important question is why the combined GW hotspots cause these changes in the Plumb flux.
Can you discuss connections, if there are any, between changes in stationary planetary wave 1 amplitude (Figures 8 and 9) and EP Flux and divergence (Figure 10)?
Figure 10: I suggest that the authors to diagnose the Transformed Eulerian Mean momentum budget in order to quantitatively understand how changes in planetary wave and gravity wave forcings lead to zonal wind changes shown in Figure 3b.
Author Response
Dear Reviewer #4,
thank you for your comments. We attached a file including a point-by-point response to your comments.
Kind regards

Reviewer 5 Report
*REVIEW OF Samtleben et al.: Mutual interference..." (atmo-964592)*
RECOMMENDATION: minor revision
** SUMMARY
The paper is devoted to the impact of regional gravity wave forcing on the circulation of the middle atmosphere. For that purpose, a couple of idealized model simulations has been run. The setup has already been used for former publications (Samtleben et al. 2019, 2020) which are correctly referenced in the paper. The new aspects are the interference between three gravity wave hotspots (Rocky Mountains, Himalayas and East Asia) which is investigated with maps and section of wind, temperature, vorticity and Plumb fluxes. The spread results are thoroughly documented and synthesized with a linearity parameter. At some points, however, the text is occasionally not exact which might cause confusion. Further, it might support the findings to add another figure illustrating the polar vortex change in strength and position - see major comment. Hence, the suggested review of the text and addition of another figure requires some effort and thats why I recommend a minor revision.
** MAJOR COMMENT
It is clear that the presentation of data from six difference fields is challenging. The authors attempt to synthesis information with the linearity parameter (eq. 1, fig. 9) which is very reasonable. In reflection on fig. 5 I suggest to present the major information in an additional figure. This figure should show the changes of the polar vortex core in a plot with the change in latitudinal position vs change in wind speed. In such a scatter plot you could place the diagnosed results from the six difference runs. The advantage would be you see immediately that, for example, HI-Ref makes the largest deceleration, while only the combination of EU+RM-Ref brings up a comparable effect in SPW1.
** TECHNICAL COMMENTS
L8: The word "preconditioning" implies a certain reason, for example the preconditioning for sudden stratospheric warmings. I suggest to use another word, for example "modification" or "change".
L11: Here I strongly suggest to add the latitudinal information because the deceleration may take place in mid-latitudes (~40 N) while acceleration in low-latitudes (~ 20 N) and, even more pronounced, in the high-latitudes (~60 N). Perhaps, it is better to write "middle latitudes" instead of "middle atmosphere" because SPW1 braking appears there (in view of Fig. 5)
L12: I see in Fig. 5 that acceleration also takes place in the higher latitudes (~60 N)?
L12: I am confused by the sentence "In contrast... mean flow." I see the increase of SPW1 amplitude at ~25 N in Fig. 5a and 8a, but I do not see the deceleration of the zonal mean flow. Instead, in Fig. 3 and 11a it is increased for lower latitudes for all simulations. Please, clarify.
L20: Insert "hotspots" after "East Asian".
L21: "arranged" --> "located"
L32: "phenomenon" --> "phenomena" (plural)
L37: "influences" --> "influence" (plural)
L53: "mostly" --> "occasionally" or "partly"
L78: "on e" --> "one"
L83 "GW hotspots" --> "GWs in hotspots"
L137: "mesospheric"? If you mean the polar vortex with its core at 55 km at 55 N, then it should be "stratospheric". The mesospheric jet is at the mesopause which is higher up and not shown.
L182: "doubled" --> "with two": Here you write "doubled" (unlcear, if in size or in number) and then you wite "three" (not tripled). So it would be more consistent writing "with one..., with two... and with three".
L189: NAO and ENSO are not the only causes of SPW changes, there is also QBO and SSW.
L197: Is the heading "Effect on the background circulation" really the best? You present below results on the zonal means and several maps. Do you mean "zonal mean" with "background"? However, I suggest to write "Effect on the circulation".
L204: If you mean "zonal mean" with "background" then it is appropriate here.
L219: "broadly" --> "roughly" / "about"
L250: May be you add "distributions" in order to differentiate from the zonal means from the section before. Perhaps, Fig. 11 could be somehow integrated in this section.
L325: Does it need a comma between "thermosphere" and "are"?
L346: "assumption" --> "hypothesis"
L347: Around 40 N? Because it is different at other latitudes.
L425: "here analyse" --> "analyse here"
L441: In Fig. 8c I see positive values also below 30 km.
L576: Well, in my perception the effect of all (Fig. 10g) is most similar to HI alone (Fig. 10c), or? EA (Fig. 10b) does not have its intensity on 30-60 km and the maxima on 70 km at 40 and 70 N are missing, too.
L734: "brewer-dobson" --> "Brewer-Dobson"
L793: "gloable" --> "global"
Fig. 5: Add description of colored boxes for RM, EA and HI in the caption.
Fig. 11: Why is it in the appendix? It could also be put in section 3.1.
** REFERENCES
Samtleben, N., C. Jacobi, P. Pišoft, P. Šácha & A. Kuchař, 2019: Effect of latitudinally displaced gravity wave forcing in the lower stratosphere on the polar vortex stability. Ann. Geopys. 37, 4: 507-523, doi:10.5194/angeo-37-507-2019.
Samtleben, N., A. Kuchař, P. Šácha, P. Pišoft & C. Jacobi, 2020: Impact of local gravity wave forcing in the lower stratosphere on the polar vortex stability: effect of longitudinal displacement. Ann. Geopys. 38, 1: 95-108, doi:10.5194/angeo-38-95-2020.
Author Response
Dear Reviewer #5,
thank you for your comments. We attached a file including a point-by-point response to your comments.
Kind regards

Round 2
Reviewer 4 Report
The authors have addressed my concerns. I recommend publication of the manuscript.